DATA RELEASE

# Citizen science data on urban forageable plants: a case study in Brazil

Filipi Miranda Soares[1,2,*], Luís Ferreira Pires[2], Maria Carolina Garcia[3], Lidio Coradin[4], Natalia Pirani Ghilardi-Lopes[5], Rubens Rangel Silva[6], Aline Martins de Carvalho[7], Anand Gavai[8], Yamine Bouzembrak[9], Benildes Coura Moreira dos Santos Maculan[10,11], Sheina Koffler[12], Uiara Bandineli Montedo[1], Debora Pignatari Drucker[13], Raquel Santiago[14], Maria Clara Peres de Carvalho[15], Ana Carolina da Silva Lima[11], Hillary Dandara Elias Gabriel[1], Stephanie Gabriele Mendonça de França[1], Karoline Reis de Almeida[1], Bárbara Junqueira dos Santos[16] and Antonio Mauro Saraiva[1]

1 Escola Politécnica, Universidade de São Paulo, São Paulo, SP, 05508-010, Brazil
2 Faculty of Electrical Engineering, Mathematics and Computer Science, University of Twente, Enschede, 7522 NB, Netherlands
3 Programa de Pós Graduação em Arquitetura, Urbanismo e Design, Centro Universitário Belas Artes de São Paulo, São Paulo, SP, 04018-010, Brazil
4 Plants for the Future Project, Brasília, DF, 70772-090, Brazil
5 Centro de Ciências Naturais e Humanas, Universidade Federal do ABC, São Bernardo do Campo, SP, 09606-045, Brazil
6 Centro Universitário Una, Belo Horizonte, MG, 30160-011, Brazil
7 Departamento de Nutrição, Faculdade de Saúde Pública, Universidade de São Paulo, São Paulo, SP, 01246-904, Brazil
8 Faculty of Behavioural, Management and Social Sciences (BMS), Industrial Engineering & Business Information Systems (IEBIS), University of Twente, Enschede, 7522 NB, Netherlands
9 Information Technology Group, Wageningen University and Research, Wageningen, 6700 HB, Netherlands
10 Programa de Pós-Graduação em Gestão & Organização do Conhecimento, Universidade Federal de Minas Gerais, Belo Horizonte, MG, 31270-901, Brazil
11 Escola de Ciência da Informação, Universidade Federal de Minas Gerais, Belo Horizonte, MG, 31270-901, Brazil
12 Instituto de Estudos Avançados, Universidade de São Paulo, São Paulo, SP, 05508-060, Brazil
13 Embrapa Agricultura Digital, Campinas, SP, 13083-886, Brazil
14 Faculdade de Nutrição, Universidade Federal de Goiás, Goiânia, GO, 74605-080, Brazil
15 Ciências e Humanidades, Universidade de São Paulo, Escola de Artes, São Paulo, SP, 03828-000, Brazil
16 Instituto de Pesquisas Energéticas e Nucleares, Universidade de São Paulo, São Paulo, SP, 05508-000, Brazil

Submitted: 28 September 2023

* Corresponding author. E-mail: filipisoares@usp.br; f.mirandasoares@utwente.nl

Preprint submitted at https://doi.org/10.1101/2024.01.22.575882

## ABSTRACT

This paper presents two key data sets derived from the *Pomar Urbano* project. The first data set is a comprehensive catalog of edible fruit-bearing plant species, native or introduced to Brazil. The second data set, sourced from the iNaturalist platform, tracks the distribution and monitoring of these plants within urban landscapes across Brazil. The study includes data from the capitals of all 27 federative units of Brazil, focusing on the ten cities that contributed the most observations as of August 2023. The research emphasizes the significance of citizen science in urban biodiversity monitoring and its potential to contribute to various fields, including food and nutrition, creative industry, study of plant phenology, and machine learning applications.

We expect the data sets presented in this paper to serve as resources for further studies in urban foraging, food security, cultural ecosystem services, and environmental sustainability.

**Subjects** Animal and Plant Sciences, Botany, Ecology

## DATA DESCRIPTION

This paper introduces two data sets related to Brazilian fruit-bearing plants. Data Set 1 presents a taxonomic list of these plant species, both native and introduced, which includes comprehensive scientific names of plants with human-edible fruits. This data set contains Scientific names (at species and family levels), Vernacular names in Portuguese, Establishment means (native or introduced), Higher geography (geographic distribution of the species in Brazil), and the last update of a given species name within the data set. A more detailed data set documentation is available on Zenodo [1].

Data Set 2, hosted on iNaturalist, compiles observations from the capitals of all 27 Brazilian federative units and specifically focuses on the species listed in Data Set 1. A backup of this data set was obtained from iNaturalist on August 22nd, 2023. The dataset features 47 columns, capturing details such as observation time, location, license, and taxonomic identification [2]. It provides an extensive taxonomic breakdown, covering classifications from kingdom and phylum down to species, subspecies, and variety (in some cases). Notably, iNaturalist allows the export of data sets annotated with Darwin Core metadata [3], ensuring data interoperability with numerous applications that accommodate data standardized with the Darwin Core standard, such as the Global Biodiversity Information Facility (GBIF).

Interoperability with GBIF is vital for several reasons. It improves the visibility and accessibility of biodiversity data worldwide, making it easier for researchers and policymakers to access this information [4]. GBIF is considered "the most extensive resource that provides access to open, integrated data on species occurrence" [5, p. 1]. This is especially important for supporting global collaborations in conservation and biodiversity research [4–6]. GBIF also ensures that data reporting is standardized and of high quality, which is crucial for reliable biodiversity studies [4, 5]. Finally, it enables more effective comparisons and cross-referencing of biodiversity data from different regions and species, enriching biodiversity research on species distribution [5].

## CONTEXT

### Urban food forests

According to the United Nations Environment Programme [7], Brazil boasts the world's richest biodiversity, accounting for 15 to 20% of the global biodiversity [7, 8]. This vast biodiversity translates into a diverse array of foods available to Brazilians. Yet, as Gomes [9] underscores, this rich biodiversity is not adequately reflected in the diets of various social groups in the country. In fact, a mere 1.3% of Brazilians actively consume biodiverse foods. Geographic location, ethnicity, age, food insecurity, gender, and educational level influence these consumption patterns [9].

The unique combination of Brazil's tropical climate and diverse ecosystems creates favorable conditions for cultivating a wide variety of fruits, many of which are native



species. These fruits are not only grown in agricultural settings but also are widely used in urban afforestation and contribute to the aesthetics of parks, squares, and sidewalks in numerous cities.

Fruit-bearing plants in urban areas, also known as urban food forests [10], may offer multiple benefits, such as enhancing food availability, particularly in areas with limited access to fresh fruit, thereby addressing food scarcity issues and improving nutritional security [11, 12]. Furthermore, they promote human health and well-being by fostering a connection with nature, providing opportunities for physical activity, and improving air quality through carbon sequestration and pollution mitigation [13].

Urban food forests also contribute to preserving and promoting biodiversity, ensuring the conservation of native fruit varieties, supporting local ecosystems, providing shelter, food, and nesting sites, and improving environmental conditions [11, 14, 15].

As explored by Hurley *et al.* [16], these green spaces contribute significantly to other non-food-related applications. Hurley and Emery [17] delved into the intricate relationship between urban forests and their utility for food, medicine, craft, and other purposes, focusing on woody species. They highlighted the material connections and diverse uses in artistic and cultural contexts of these green spaces, underscoring their multifaceted importance in urban settings [17].

By acknowledging the potential of edible green infrastructures, urban areas can mitigate the adverse impacts of urbanization while simultaneously enhancing the sustainability and resilience of cities [11, 12].

## Use of citizen science for urban biodiversity monitoring

Citizen Science (CS) is a collaborative endeavor between scientists and the public that involves the active participation of members of society in scientific research. Participating in a CS initiative to collect data on fruit-bearing plants locally can be motivated by various reasons, including curiosity and people's desire to engage in these initiatives.

CS platforms offer a conducive environment for citizen scientists to contribute to science and learn. iNaturalist, for instance, serves as an online social network that supports a community of users that exchange biodiversity information, fostering collaborative learning about the natural world [18]. It also offers both a mobile and a web-based platform that allows users to upload photos to be analyzed by Artificial Intelligence, which provides organism identification based on these photos and other variables, such as the location of the observation.

The wealth of data available on iNaturalist can be leveraged for many research initiatives that center around monitoring urban biodiversity. Some notable studies have utilized iNaturalist to track and understand the presence of alien species. For example, in the Tyumen Region, Russia, researchers [19] compared data from the entire region with the Tyumen Urban Area to assess the distribution of four alien plant species. Additionally, in Kyiv city, the expansion of the invasive Balkan slug (*Tandonia kusceri*) in urban landscapes was studied [20].

Insects (class Insecta) have also been a focal point of research in urban environments using iNaturalist. Studies have explored various aspects of insect biodiversity, such as the species richness of butterflies (Lepidoptera: Rhopalocera) in Los Angeles [21], the compilation of a species checklist of hawkmoths (Lepidoptera) in Baguio City, Philippines [22], and the monitoring of Eastern Carpenter Bees (*Xylocopa virginica*) in urban areas of



the United States and Canada to detect wing pigment loss [23]. CS projects using iNaturalist have also targeted other animal groups, including mammals, birds, amphibians, and reptiles. For instance, studies monitored the presence of large mammals in urban centers in North America during the COVID-19 pandemic [24]. Urban red foxes and coyotes were monitored using iNaturalist observations in North America [25, 26]. Bird sightings in the city of Chicago, Illinois, were documented through iNaturalist [27]. Moreover, research characterized amphibians' and reptile species' richness in ecological recreational parks in Mérida, Yucatán, Mexico [28], and in the Edith L. Moore Nature Sanctuary in Houston, Texas [29].

Several studies with broader objectives have also leveraged iNaturalist data to understand ecological interactions in cities [30–32], monitor urban biodiversity [33, 34], explore human use of urban green areas in Europe [35], and examine insect management practices for urban farming [36].

This range of research studies demonstrates iNaturalist's versatility and potential as a tool for exploring urban biodiversity and ecological dynamics. It also indicates the platform's potential application in creating open-source data bases on forageable plants, as explored by [37].

## METHODS

### Data set 1

Version 3 of the list of native and exotic fruit-bearing plant species cultivated in Brazil was built upon two main sources based on specialists' recommendations: the book *Brazilian Fruits and Cultivated Exotics* [38] and the book series *Plants for the Future.* The latter includes volumes specifically dedicated to species of economic value from four regions of Brazil, namely the South [39], Midwest [40], Northeast [41], and North [8]. A botanist carefully reviewed this list in the following steps:

(i)   Scientific Name Verification. The primary task involved checking each scientific name to ensure the species name's existence, correctness, and validity. The review focused on determining if the species' names were accepted or synonyms at the current time. Key resources for this verification included the Flora and Fungi of Brazil catalog [42], Tropicos.org by the Missouri Botanical Garden [43], the International Plant Names Index [44], Kew Botanic Gardens Plants of the World Online [45], and the New York Botanical Garden Vascular Plant Types Catalog [46]. It is important to note that scientific names are dynamic; a species considered valid today might become a synonym tomorrow, and vice versa. Thus, we will continuously update this list.

(ii)   Origin Check. A thorough check was conducted for each species to determine whether it was native or introduced. This involved extensive online research and consultation of relevant literature, primarily focusing on the *Plants for the Future* book series [8, 39–41].

(iii)   Two new columns were added to Version 3 of the list: Botanical Family and Geographical Distribution. Information for these columns was primarily sourced from the Flora and Fungi of Brazil catalog and Tropicos.org.

(iv)   Handling Synonyms. Many species included in Version 2 of the list from the reference books were identified as synonyms according to the consulted online catalogs. In such cases, the currently valid scientific names replaced the synonyms.

(v)   Removed species. In Version 3 of the list, some plant species from Version 2 were excluded due to their lack of importance as food sources.

(vi) Inclusion of Additional Species. Some fruit-bearing species listed in the *Plants for the Future* books [8, 39–41] that were missing in Version 2 were included in Version 3.

## Data set 2

iNaturalist provides tools for managing observations of interest, known as Projects [47]. We used two types of projects:

- Collection Projects function as a saved database search with enhanced features for data visualization, such as a distinctive banner, icon, URL, and a journal that enables communication with project followers [47]. When creating a Collection project, specific requirements can be set, such as "taxa, place(s), users, dates, and quality grade". Each time the project's page is accessed, "iNaturalist will perform a quick search and display all observations that match the project's requirements" [47].
- Umbrella Projects are designed to compile, compare, or promote multiple collection projects [47].

The criteria for the *Pomar Urbano* collection projects were established as follows: user-registered observations must be about fruit-bearing plant species listed in the taxonomic compilation of Data Set 1. Furthermore, each observation should originate from one of the capitals of Brazil's 27 federative units, such as São Paulo, Rio de Janeiro, or Brasília. To compile all this data, the umbrella project has been set up to aggregate data from all 27 corresponding collection projects [48].

## DATA SAMPLE

Version 3 of Data Set 1 counts 429 fruit-bearing species. Notably, 398 of these species have been observed by iNaturalist users in various locations across Brazil as of 22nd August 2023. *Pomar Urbano* has amassed a collection of over 10,943 observations throughout Brazil, referred to in this paper as Data Set 2. This data set was generated both from opportunistic records on iNaturalist, without the use of active engagement strategies by the project coordinators, and biological surveying (bioblitz).

Figure 1 shows the top 10 cities with the highest number of observations. This ranking spans all five macro-regions of Brazil. Brasília (Midwest region) leads this ranking, followed by São Paulo and Rio de Janeiro (both in the Southeast region).

Figure 2 depicts some observations from Brasília. Among the ten most frequently observed species, seven are native to Brazil, namely *Caryocar brasiliense, Eugenia dysenterica, Psidium guajava, Eugenia uniflora, Bixa orellana, Hymenaea courbaril,* and *Anacardium humile*. The species with the highest number of observations is *Caryocar brasiliense*, which is endemic to the Brazilian Cerrado biome, encompassing the entire territory of Brasília [49]. This particular species holds significant cultural importance in Midwest Brazilian Cuisine according to [50], potentially accounting for its widespread presence and dissemination throughout Brasília.

The city of São Paulo presents a distinct scenario (see Figure 3). Among the most observed species in the city, only four of them (*Eugenia uniflora, Syagrus romanzoffiana, Plinia cauliflora,* and *Psidium guajava*) are native to Brazil. This highlights a significant difference in species composition compared to Brasília.

Furthermore, there were notable variations when considering the species that garnered the highest number of observations in each city. Only *Psidium guajava, Eugenia uniflora,*



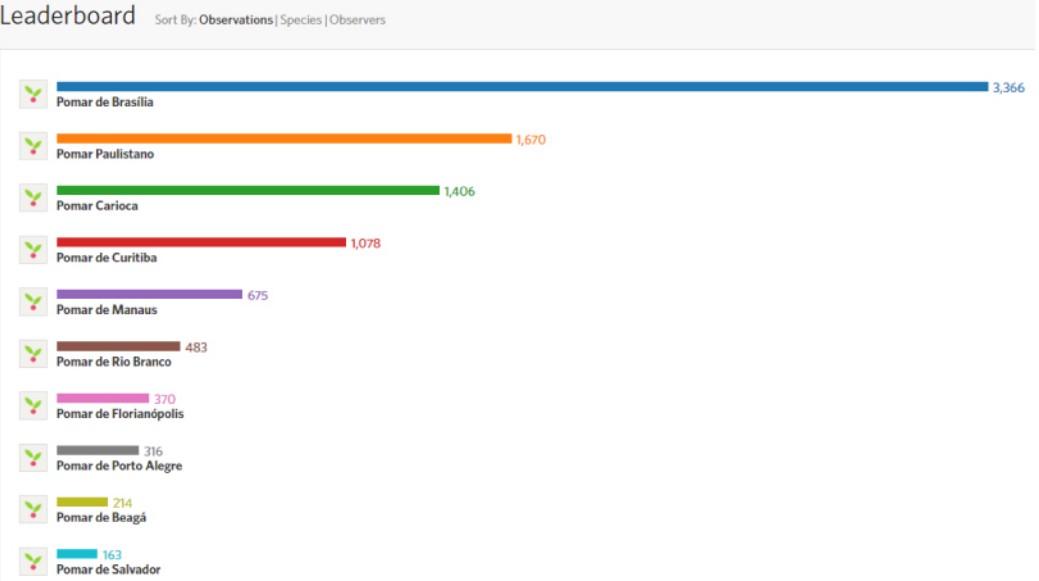

**Figure 1.  Ten Brazilian cities with the highest number of observations of fruit-bearing species.**
Note: Screenshot from the *"Pomar Urbano"* project on iNaturalist, based on observations collected as of August 2023. This image represents the data available at that time and may not reflect the most recent updates or additions. To view the most up-to-date data and additional contributions, visit the project online at: https://www.inaturalist.org/projects/pomar-urbano.

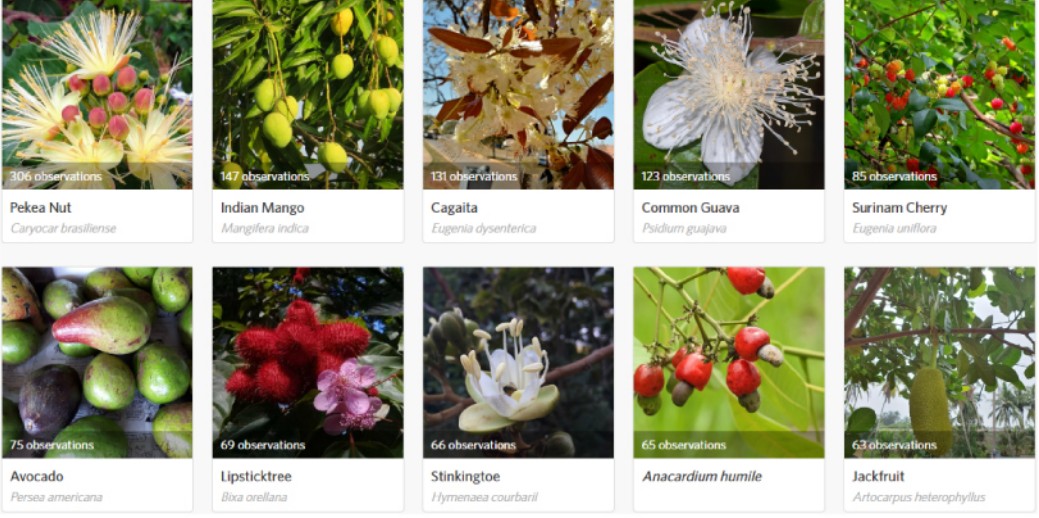

**Figure 2.  Ten most observed species in Brasília.**
Note: Screenshot from the *"Pomar de Brasília"* project on iNaturalist, based on observations collected as of August 2023. This image represents the data available at that time and may not reflect the most recent updates or additions. To view the most up-to-date data and additional contributions, visit the project online at: https://www.inaturalist.org/projects/pomar-de-brasilia.

and *Persea americana* emerged within the most observed species in both Brasília and São Paulo, since these cities are situated in regions characterized by distinct biomes, differences in species diversity between these locations were expected. The contrasting



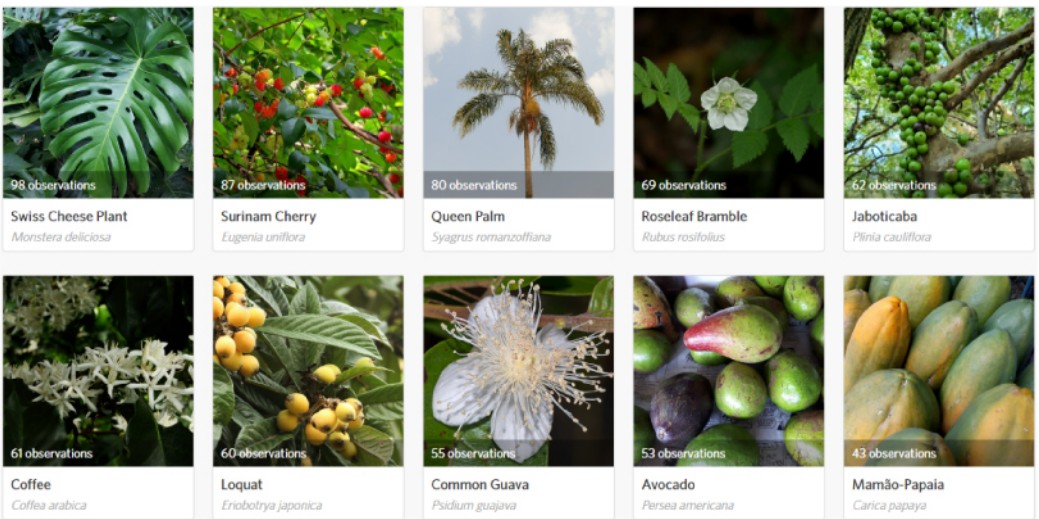

**Figure 3.** Ten most observed species in São Paulo.
Note 1: Screenshot from the "*Pomar Paulistano*" project on iNaturalist, based on observations collected as of August 2023. This image represents the data available at that time and may not reflect the most recent updates or additions. To view the most up-to-date data and additional contributions, visit the project online at: https://www.inaturalist.org/projects/pomar-paulistano. Note 2: Recent taxonomic advancements have led to the synonymization of *Plinia cauliflora* with *Plinia peruviana*, with the latter now being the preferred nomenclature, as detailed in the Flora and Funga of Brazil [42]. However, at the time of this article's publication, this taxonomic update had not been reflected on iNaturalist, resulting in the figure retaining the previous name, *Plinia cauliflora*.

biomes contribute to the variation in the composition and distribution of fruit-bearing species, highlighting the significance of considering regional ecological factors when analyzing biodiversity patterns.

In Rio de Janeiro (see Figure 4), the abundance of native fruit-bearing species is also evident. Among the ten most observed species, a noteworthy seven of them are native to Brazil, including *Eugenia uniflora, Syagrus romanzoffiana, Lecythis pisonis, Myrciaria glazioviana, Pachira aquatica, Opuntia monacantha*, and *Eugenia brasiliensis*.

Both *Eugenia uniflora* and *Syagrus romanzoffiana* emerge as prominent species within the top ten observations across all three cities. The remarkable prevalence of *Eugenia uniflora* can be attributed to its adaptability to diverse climate and soil conditions [51]. This adaptability, coupled with the increasing economic interest in the berries produced by this plant, has led to its cultivation in various regions worldwide, not just in Brazil [51].

The significant presence of *Syagrus romanzoffiana* in the three cities can be attributed to its adaptability to the biomes prevalent in these regions, including the Atlantic Forest and Cerrado [52]. This species' fruit pulp and nut can be consumed by humans and also holds great value for various groups of animals, indicating its ecological importance in providing food resources [52, 53]. In addition, native communities believe that this palm species is a sacred plant, attributed with cosmogonic and cosmological spiritual significance [53]. Furthermore, recent studies have shed light on the medicinal potential of *Syagrus romanzoffiana*, demonstrating its effectiveness in treating infections and chronic diseases [54]. Additionally, the seeds of this plant have been found to possess a high concentration of oil, making them a promising candidate for biodiesel production [55].

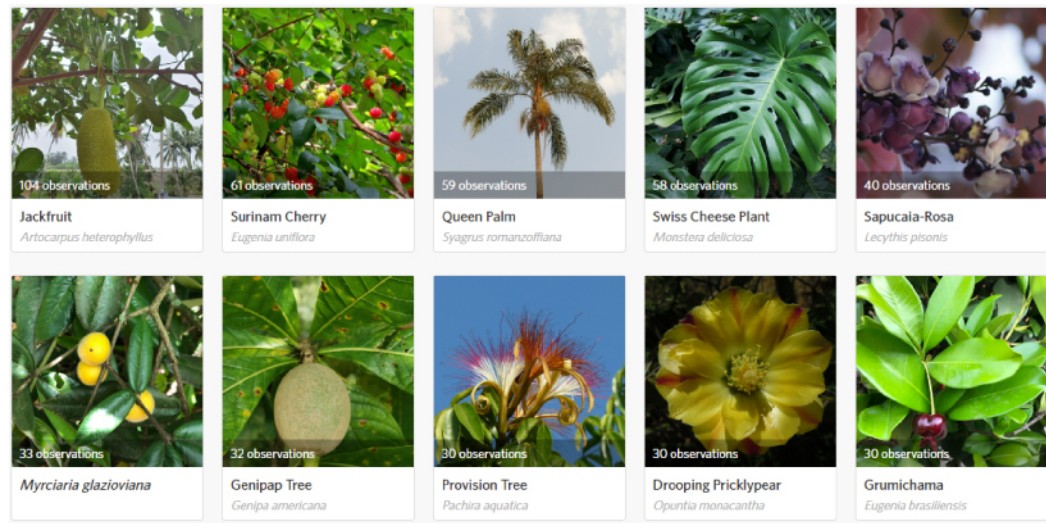

**Figure 4.   Ten most observed species in Rio de Janeiro.**
Note: Screenshot from the "*Pomar Carioca*" project on iNaturalist, based on observations collected as of August 2023. This image represents the data available at that time and may not reflect the most recent updates or additions. To view the most up-to-date data and additional contributions, visit the project online at: https://www.inaturalist.org/projects/pomar-carioca.

## DATA VALIDATION AND QUALITY CONTROL

Species identification on iNaturalist leverages a Computer Vision Model (CVM) to process observation images and suggest potential species names [56–58]. Although the CVM lacks data on all 2 million species registered worldwide, it successfully identifies about 85% of observations [57, 58]. For species not recognized by the CVM, it recommends higher taxonomic classifications like genus or family [58].

A key strength of iNaturalist is its community of specialists, known as identifiers, who significantly enhance the platform's species identification accuracy [56]. These identifiers also contribute to the CVM by providing quality data and validating observations' identifications [56]. Their expertise is particularly valuable in distinguishing between similar species, thereby refining the CVM and improving its suggestions over time [56, 58].

The ongoing contributions of these identifiers are vital for the CVM's continuous improvement. By providing a high-quality dataset and expanding the model's species coverage, they significantly boost its overall performance. For more insights on the impact of identifiers on iNaturalist data quality, refer to [56].

iNaturalist enforces data quality criteria, classifying records into various levels. The highest is "Research Grade" (RG), characterized by specific criteria including having a valid date, location, photo or sound, being of a non-captive or non-cultivated organism, and community consensus on the organism identification at the species level or lower [59]. GBIF indexes RG observations, making them accessible for download [60].

In the *Pomar Urbano* project, prioritizing RG data from iNaturalist would be logical due to the necessity for accurate plant species identification, especially when health risks from non-edible fruits are concerned. However, a significant challenge arises since the project focuses on forageable plants, many of which are cultivated and thus are not eligible for RG status on iNaturalist.

This categorization is particularly complex in urban settings where the line between cultivated and wild plants can blur. For instance, a plant initially cultivated by humans might become naturalized in an urban environment, raising questions about its categorization. This ambiguity is evident in discussions within the iNaturalist community, where there is an ongoing debate about how to handle observations of plants that might have been planted initially but have become self-sustaining and naturalized in an urban context. For further insight into the complexities surrounding the classification of plants as "cultivated" in urban contexts, refer to the iNaturalist forum discussion [61].

Given these nuances, relying exclusively on RG data in the *Pomar Urbano* project could be limiting. The project's focus on urban environments means many valuable observations might be of cultivated plants. Therefore, while RG observations are preferred for their verified accuracy, the project would benefit from a more inclusive approach, considering data also from non-RG observations. Still, we believe this data set could also be relevant to GBIF and other biodiversity data platforms, considering the potential interest in data that sheds light on the distribution and characteristics of economically important plant species within urban ecosystems.

## REUSE POTENTIAL

The data sets presented in this paper provide a foundational platform for understanding the diversity of fruit-bearing plants in select Brazilian cities. We assert that this contributes to various open research questions in the existing literature on urban foraging and ecosystem services in urban environments, including, but not limited to:

- Examining how the current species composition in cities forms pre-existing landscapes that are both edible and useful, as explored in the works of [16, 17, 62];
- Developing open data sources focused on Urban Food Forests, as highlighted by [37];
- Investigating the role of urban forests in supporting cultural ecosystem services, as discussed by [63];
- Analyzing urban foraging as a means to enhance food security [64–69];
- Facilitating the sharing of ecological knowledge, as mentioned in [70];
- Contributing to social-ecological resilience, as studied by [71];
- Aiding in reconnecting urban residents with nature and biodiversity, as described by [72–74].

In the context of the Pomar Urbano project, data reuse includes:

## Food and nutrition

To engage and inspire individuals to explore the culinary potential of locally available fruit, we partnered up with the food laboratory Sustentarea at the University of São Paulo. In this facility, we document food recipes showcasing the most frequently encountered fruit in São Paulo, Brazil. Additionally, we have conducted parallel experiments in Brasília. These recipes will be shared across social media platforms to encourage people to prepare easy, affordable, tasty dishes using these regional ingredients. Figure 5 shows the results of some of these experiments.

As our initiative progresses, we envision expanding our recipe collection to include fruit from other cities. The book *Biodiversidade Brasileira: sabores e aromas* (Brazilian Biodiversity: flavors and aromas) [75] is also being used as reference for preparing Brazilian recipes using fruit.

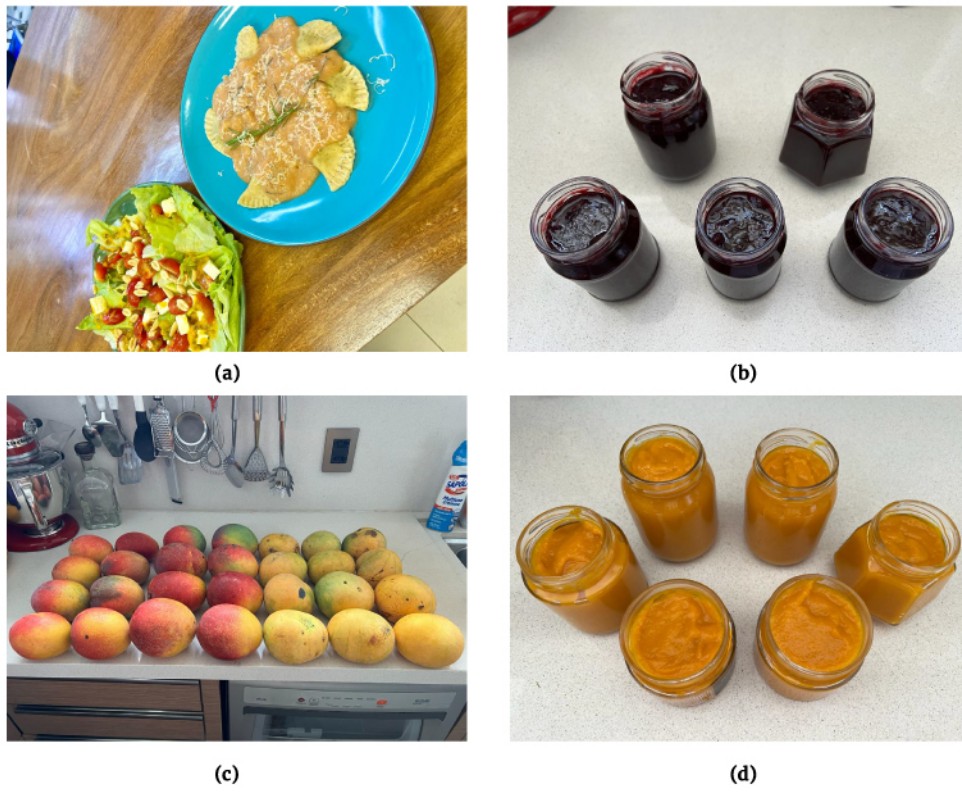

**Figure 5.** **Recipes prepared for *Pomar Urbano*.**
(a) Green salad with passion fruit (*Passiflora sp.*) sauce and a taioba (*Xanthosoma taioba*) tortelli with guava (*Psidium guajava*) sauce. Recipe production: Ana Maria Bertolini, Gabriela Rigote, Natalie Marinho Dantas. Photo: Gabriela Rigote [48]. (b) Jaboticaba (*Plinia peruviana*) jam, prepared in December 2023 in Brasília, Brazil. Recipe preparation and Photo: Lidio Coradin and Vera Coradin. (c) Freshly foraged mangoes, Brasília, Brazil, December 2023. This photograph shows ripe mangoes (*Mangifera indica*), handpicked during a foraging expedition. Photo: Lidio Coradin. (d) Mango chutney prepared by Lidio Coradin and Vera Coradin with foraged mango in Brasília, Brazil, December 2023. Photo: Lidio Coradin.

## Creative industry

The creative industry, also known as creative economy, encompasses a diverse range of sectors dedicated to generating and commercializing creativity, art, and original content. This industry spans advertising, architecture, arts and crafts, design (including fashion), film, music, performing arts, publishing, and software (like video games) [76]. It thrives on innovation and caters to cultural and economic demands, often blurring the lines between art and commerce. The creative industry may benefit from our data in many ways.

We are identifying Brazilian digital influencers on prominent platforms such as TikTok and Instagram, focusing on those who delve into topics like sustainable diets, vegetarianism, science, nature, and related areas. We aim to collaborate with these influencers, seeking their endorsement to introduce our initiative to their followers. By leveraging data from our project, these influencers can generate content while we gain the advantage of increased user participation in our data collection efforts.

Beyond content creation, information on fruit-bearing plants can inspire the development of diverse products. Figure 6 showcases products crafted by Brazilian artists inspired by observations recorded on *Pomar Urbano*.

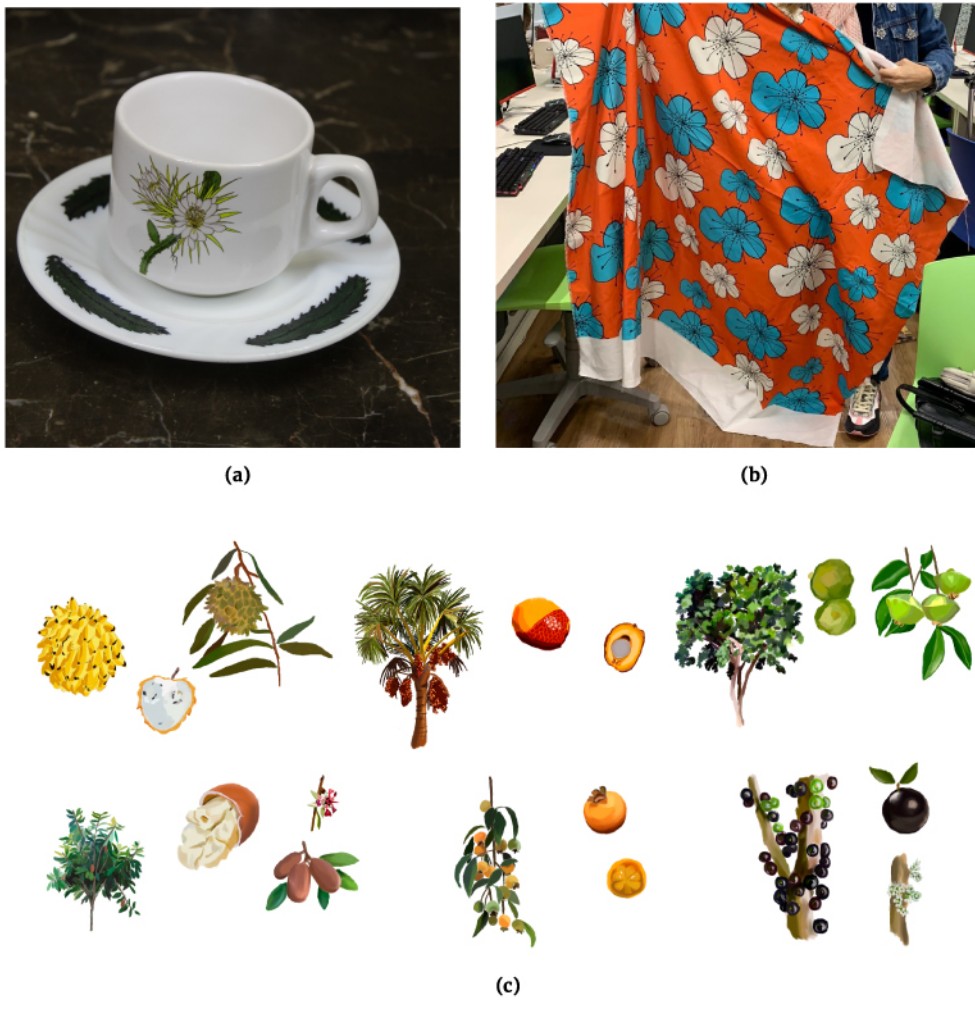

**Figure 6. Product designs inspired by *Pomar Urbano*.**
(a) A porcelain cup featuring a print inspired by the Night Blooming Cactus flower (*Epiphyllum oxypetalum*), observed during a research survey in São Paulo, Brazil. The design was created by Fashion Design students Kelly Cristina Soares Barbieri, Larissa Galdino de Souza Costa, and Karollina Brandão Araújo Cosso at Centro Universitário Belas Artes de São Paulo, supervised by Maria Carolina Garcia [48]. (b) Print for a tablecloth inspired by the guava tree flower (*Psidium guajava*), using the traditional technique of Brazilian Chita. Created by Luciana Mendonça, a student of Interior Design at the Centro Universitário Belas Artes de São Paulo under the supervision of Maria Carolina Garcia [48]. (c) Digital illustrations showcasing a diverse array of Brazilian plants (from the up left corner to the right down corner: *Annona mucosa, Mauritia flexuosa, Campomanesia phaea, Theobroma grandiflorum, Campomanesia xanthocarpa,* and *Plinia peruviana*). Designed for the Pomar Urbano project based on multiple photos retrieved from iNaturalist.org. Illustrations: Bruna Stefani Perin and Fernanda Beatriz Fernandez Correa. Supervision: Rubens Rangel Silva (Ânima Educação).

## Understanding plant phenology

Numerous reference sources offer information on the timing of fruiting and flowering for various species in Brazil. Notably, the work conducted by Coradin, Camillo, and Vieira [8] stands out, presenting an extensive table that details the aforementioned plant phenology aspects, including their respective peaks, for species found in the North of Brazil. Drawing from these existing literature resources, we anticipate harnessing the provided data to support our future analyses concerning the fruiting seasons of each species.

Our primary objective is to compare the data collected by citizen scientists with the findings presented by reference sources. Through this comparative analysis, we aim to ascertain whether the information gathered through CS initiatives yields similar outcomes to those achieved by professional scientists. This investigation aims to assess the accuracy and reliability of CS data collection within the specific framework of our project.

By systematically examining and contrasting these results, we can shed light on the efficacy and potential of CS as a means of data collection for such projects. This comparison will serve as a crucial step towards enhancing our understanding of seasonal fruit patterns while also showcasing the contributions that citizen scientists can make in advancing scientific knowledge. Furthermore, this data can potentially highlight temporal shifts in the reproductive seasonality of the monitored plant species in the long term.

## Machine learning applications

The safety and authenticity of food can be safeguarded through data derived from fruit-bearing plants, serving as a foundational reference for genuine fruit attributes. Machine Learning (ML) algorithms can then be employed to pinpoint irregularities in fruit quality or composition, thereby enabling the detection of deceptive practices and potential food safety hazards [77]. Furthermore, within urban settings, the local production of fruits and vegetables can be seamlessly integrated into the urban supply chain to enhance supply chain management. ML algorithms can optimize transportation logistics and minimize food wastage, leading to a more sustainable and efficient urban food system. Moreover, in the face of environmental challenges, ML offers a powerful tool to analyze long-term data from urban fruit crops. This data-driven approach can provide valuable insights into understanding the impact of climate change and urbanization on plant growth and biodiversity. Such knowledge can be pivotal in guiding urban planning decisions and conservation initiatives.

In conclusion, when coupled with ML capabilities, the wealth of data produced by fruit-bearing plants in urban environments can significantly impact issues related to food authenticity, safety, and environmental concerns. These applications enhance the quality of urban life and contribute to global endeavors to build sustainable and resilient urban ecosystems.

For additional insights into citizen science for urban foraging, refer to the related commentary paper in GigaScience [78].

## DATA AVAILABILITY

The datasets are available on Zenodo: Data Set [1] and Data Set [2]. Supporting data is also available via the GigaScience database, GigaDB [79].

## LIST OF ABBREVIATIONS

AI: Artificial Intelligence; CS: Citizen Science; CVM: Computer Vision Model; RG: Research Grade.

## DECLARATIONS

## Competing interests

The authors declare that they have no competing interest regarding the publication of this work. There are no financial, personal, or professional relationships that could be perceived as potentially biasing the content presented in this manuscript.

## Authors' contributions

Study Conceptualization: FMS, MCG, AMS, LFP, BCMSM. Data curation: FMS, LC, MCPC, ACSL, SGMF, HDEG, BJS. Formal Analysis: FMS, LC, LFP, RRS, NPGL. Funding acquisition: AMS, UBM, BCMSM, MCG, RRS. Investigation: FMS, LFP, MCG, LC, NPGL, RRS, AMC, BCMSM, SK, UBM, DPD, RS, AMS. Methodology: FMS, NPGL, SK, UBM. Project administration: FMS, AMS. Supervision: AMS, LFP, BCMSM, UBM, AMC, MCG, RRS. Validation: LC. Data Visualization: FMS, RRS. Writing – original draft: FMS, LFP, LC, NPGL, AG, YB, SK. Writing – review & editing: All authors made significant contributions to the review and editing of this manuscript.

## Funding

FMS thanks the Fundação de Amparo à Pesquisa do Estado de São Paulo (FAPESP) (Process number: 21/15125-0, and 22/08385-8). BCMSM thanks the Conselho Nacional de Desenvolvimento Científico e Tecnológico (CNPq) (Process number: 303650/2019-2). AMS, NPGL, SK, and FMS thank FAPESP (Process number: 2018/14994-1). SK thanks FAPESP (Process number: 2019/26760-8).

## Acknowledgements

We express our gratitude to the more than 2,600 citizen scientists who have made valuable contributions through their observations.

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
