## [Reviewer Report]

Comments on revised manuscriptI appreciate the opportunity to have another look at this manuscript. Indeed, I commend the authors for addressing Reviewer 1 and Reviewer 2 comments quite thoroughly. I also think the balance on more iNat/GBIF stuff by me, plus more urban forageable plants by the second Reviewer has done a good job of strengthening this data paper. Well done!

---

## [Editor Report]

Editor’s AssessmentThis is a Data Release paper describing data sets derived from the Pomar Urbano project cataloging edible fruit-bearing plants in Brazil. Including data sourced from the citizen science iNaturalist app, tracking the distribution and monitoring of these plants within urban landscapes (Brazilian state capitals). The data was audited and peer reviewed and put into better context, and there is a companion commentary in GigaScience journal better explaining the rationale for the study. Demonstrating this data providing a platform for understanding the diversity of fruit-bearing plants in select Brazilian cities and contributing to many open research questions in the existing literature on urban foraging and ecosystem services in urban environments.

---

## [Reviewer Report]

Reviewer name and names of any other individual's who aided in reviewer Corey T. CallaghanDo you understand and agree to our policy of having open and named reviews, and having your review included with the published papers. (If no, please inform the editor that you cannot review this manuscript.)YesIs the language of sufficient quality?YesPlease add additional comments on language quality to clarify if needed
Are all data available and do they match the descriptions in the paper? YesAdditional CommentsAre the data and metadata consistent with relevant minimum information or reporting standards? See GigaDB checklists for examples <a href="http://gigadb.org/site/guide" target="_blank">http://gigadb.org/site/guide</a>YesAdditional CommentsMore information should be given on the relevance to GBIF. And why the dataset is necessary to 'stand alone'. The main reason I guess is because in this context cultivated organisms are really valuable as a lot of your target organisms will indeed be cultivated.Is the data acquisition clear, complete and methodologically sound?NoAdditional CommentsMore detail should be provided about the difference in research grade and cultivated organisms on iNaturalist. The RG could be downloaded from GBIF, but I understand the need to go around that given that the cultivated organisms are also valuable in this context.Is there sufficient detail in the methods and data-processing steps to allow reproduction?YesAdditional CommentsIs there sufficient data validation and statistical analyses of data quality? YesAdditional CommentsNot relevant.Is the validation suitable for this type of data?NoAdditional CommentsThere should be more information provided on the CV model. And more information provided on the importance of identifiers in iNaturalist ecosystem. They are critically important. Right now, it reads as if the CV model generally accurately identifies organisms, but this isn't necessarily true, and there is no reference given. However, the identifiers are necessary to help data processing and identification of the organisms submitted to iNaturalist. I also think the biases of cultivated organisms not being identified as readily by iNaturalist identifiers should be discussed somewhere in the manuscript.Is there sufficient information for others to reuse this dataset or integrate it with other data?YesAdditional CommentsYes.Any Additional Overall Comments to the AuthorI appreciated the description of this dataset and particularly liked the 'context' section and think it did a good job of setting up the need for such data. I would use iNaturalist throughout as opposed to iNat since iNat is a bit more colloquial.RecommendationMinor Revision

---

## [Reviewer Report]

Reviewer name and names of any other individual's who aided in reviewer Patrick HurleyDo you understand and agree to our policy of having open and named reviews, and having your review included with the published papers. (If no, please inform the editor that you cannot review this manuscript.)YesIs the language of sufficient quality?YesPlease add additional comments on language quality to clarify if needed
Are all data available and do they match the descriptions in the paper? YesAdditional CommentsAre the data and metadata consistent with relevant minimum information or reporting standards? See GigaDB checklists for examples <a href="http://gigadb.org/site/guide" target="_blank">http://gigadb.org/site/guide</a>YesAdditional CommentsIs the data acquisition clear, complete and methodologically sound?YesAdditional CommentsIs there sufficient detail in the methods and data-processing steps to allow reproduction?YesAdditional CommentsIs there sufficient data validation and statistical analyses of data quality? YesAdditional CommentsIs the validation suitable for this type of data?YesAdditional CommentsIs there sufficient information for others to reuse this dataset or integrate it with other data?YesAdditional CommentsAny Additional Overall Comments to the AuthorThis is a very interesting paper and approach to examining questions related to the presence of edible plants in Brazilian cities. As such, it addresses--whether intentionally or not--open questions within the existing literatures of urban foraging and urban ecosystem services (Shackleton et al. 2017, ), among others, including:   1. how the existing species composition of cities create already existing edible/useful landscapes (see Hurley et al. 2015, Hurley and  Emery 2018, Hurley et al. 2022), or what the authors appear to describe as "orchards", and including the use of open data sources to  support these activities (Stark et al. 2019),  2. the ways that urban forests support cultural ecosystem services (Plieininger et al. 2015),  2a. dietary need/food security (Synk et al. 2017, Bunge et al. 2019, Gaither et al. 2020, Sardeshpande & Shackleton 2023), including in  Brazil (Brito et al 2020), and diversity (Gareake & Shackleton 2020),  2b. sharing of ecological knowledge (Landor-Yamagata 2018), and  2c. social-ecological resilience (Sardeshpande et al. 2021) as well as  2d. reconnect urban residents to nature/biodiversity (Palliwoda et al. 2017, Fisher and Kowarik 2020, Schunko and Brandner 2022).   3. I note that while most of the literatures above focus on foods and edibility, Hurley et al. 2015 and Hurley and Emery consider the relationship of urban forests for other, not food-related uses and thus the material connections and uses by people within art and other cultural objects.   4. I also note that some scholars are beginning to focus on the question of urban governance and the inclusion of urban fruit trees (Kowalski & Conway 2023), building off of the rapidly expanding literature on urban food forestry (Clark and Nicholas 2011) and edible green infrastructure. The difference between these literatures and those I've suggested above is that they generally focus on policy and planting interventions to insert, add, or otherwise enhance the edibility of these spaces (as opposed to the above stream analyzing how people interact with what is already there, whether those species are intended for harvest by people, or not, and thus it seems like this piece better links to those issues .  5. It would be helpful to see at least some of these links between the present research and its focus on methods for using a particularly valuable dataset linked to/with efforts to address the conceptual questions that are raised by the authors. For example, in relation to item #1 above, I might suggest dropping the use of "orchard" and describe the species being analyzed as representative of an "actually existing food forests" within these cities (building on the existing literature Items 1 through 3), while indicating the insights it might provide to those interested in interventions to shape future cities and their species composition to enhance human benefits (items 4 and 5). Likewise, it would be helpful to reference the items in 2a through 2d where they appear in the Context section, building on the very high level citations already (e.g., current citations #5 FAO and #6 Salbitano).   To be clear, much of what I'm asking for here can be, I think, addressed through additions of single sentences or phrases throughout the context section, along with brief reference to these within the brief discussions under "Reuse Potenial".  Or perhaps this is too in-depth for this journal. If that's the case, then I do think that reference to several key articles is needed, specifically to signal the insights this piece has for this ongoing work to understand how urban forests function for human benefit. Those would be:  Shackleton et al. 2017, Hurley & Emery 2018, Garekea & Shackleton 2020, Fisher & Kowarik 2020, Sardeshpande et al. 2021.   Most critically, the work of Stark et al. 2019 should be acknowledged.   My sincere thanks to the authors to learn from this work and my apologies for the delay in completing this review.   Works Cited Above  Bunge, A., Diemont, S. A., Bunge, J. A., & Harris, S. (2019). Urban foraging for food security and sovereignty: quantifying edible forest yield in Syracuse, New York using four common fruit-and nut-producing street tree species. Journal of Urban Ecology, 5(1), juy028.  Fischer, L. K., & Kowarik, I. (2020). Connecting people to biodiversity in cities of tomorrow: Is urban foraging a powerful tool?. Ecological Indicators, 112, 106087.  Garekae, H., & Shackleton, C. M. (2020). Foraging wild food in urban spaces: the contribution of wild foods to urban dietary diversity in South Africa. Sustainability, 12(2), 678.  Hurley, P. T., Emery, M. R., McLain, R., Poe, M., Grabbatin, B., & Goetcheus, C. L. (2015). Whose urban forest? The political ecology of foraging urban nontimber forest products. Sustainability in the global city: Myth and practice, 187-212.  Hurley, P. T., & Emery, M. R. (2018). Locating provisioning ecosystem services in urban forests: Forageable woody species in New York City, USA. Landscape and Urban Planning, 170, 266-275.  Hurley, P. T., Becker, S., Emery, M. R., & Detweiler, J. (2022). Estimating the alignment of tree species composition with foraging practice in Philadelphia's urban forest: Toward a rapid assessment of provisioning services. Urban Forestry & Urban Greening, 68, 127456.  Kowalski, J. M., & Conway, T. M. (2023). The routes to fruit: Governance of urban food trees in Canada. Urban Forestry & Urban Greening, 86, 128045.  Landor-Yamagata, J. L., Kowarik, I., & Fischer, L. K. (2018). Urban foraging in Berlin: People, plants and practices within the metropolitan green infrastructure. Sustainability, 10(6), 1873.  Palliwoda, J., Kowarik, I., & von der Lippe, M. (2017). Human-biodiversity interactions in urban parks: The species level matters. Landscape and Urban Planning, 157, 394-406.  Plieninger, T., Bieling, C., Fagerholm, N., Byg, A., Hartel, T., Hurley, P., ... & Huntsinger, L. (2015). The role of cultural ecosystem services in landscape management and planning. Current Opinion in Environmental Sustainability, 14, 28-33.  Sardeshpande, M., Hurley, P. T., Mollee, E., Garekae, H., Dahlberg, A. C., Emery, M. R., & Shackleton, C. (2021). How people foraging in urban greenspace can mobilize social–ecological resilience during Covid-19 and beyond. Frontiers in Sustainable Cities, 3, 686254.  Sardeshpande, M., & Shackleton, C. (2023). Fruits of the city: The nature, nurture and future of urban foraging. People and Nature, 5(1), 213-227.  Schunko, C., & Brandner, A. (2022). Urban nature at the fingertips: Investigating wild food foraging to enable nature interactions of urban dwellers. Ambio, 51(5), 1168-1178.  Shackleton, C. M., Hurley, P. T., Dahlberg, A. C., Emery, M. R., & Nagendra, H. (2017). Urban foraging: A ubiquitous human practice overlooked by urban planners, policy, and research. Sustainability, 9(10), 1884.  Stark, P. B., Miller, D., Carlson, T. J., & De Vasquez, K. R. (2019). Open-source food: Nutrition, toxicology, and availability of wild edible greens in the East Bay. PLoS One, 14(1), e0202450.  Synk, C. M., Kim, B. F., Davis, C. A., Harding, J., Rogers, V., Hurley, P. T., ... & Nachman, K. E. (2017). Gathering Baltimore’s bounty: Characterizing behaviors, motivations, and barriers of foragers in an urban ecosystem. Urban Forestry & Urban Greening, 28, 97-102. 
RecommendationMinor Revision